# Natural Killer Cell Receptors and Endometriosis: A Systematic Review

**DOI:** 10.3390/ijms24010331

**Published:** 2022-12-25

**Authors:** José Lourenço Reis, Natacha Nurdine Rosa, Miguel Ângelo-Dias, Catarina Martins, Luís Miguel Borrego, Jorge Lima

**Affiliations:** 1Department of Obstetrics and Gynecology, LUZ SAÚDE, Hospital da Luz, 1500-650 Lisboa, Portugal; 2UCD School of Medicine, University College Dublin, D04 V1W8 Dublin, Ireland; 3Comprehensive Health Research Centre (CHRC), NOVA Medical School|Faculdade de Ciências Médicas (NMS|FCM), Universidade Nova de Lisboa, 1169-056 Lisboa, Portugal; 4Immunology Department, NOVA Medical School|Faculdade de Ciências Médicas (NMS|FCM), Universidade Nova de Lisboa, 1169-056 Lisboa, Portugal; 5Department of Imunoallergy, LUZ SAÚDE, Hospital da Luz, 1500-650 Lisboa, Portugal

**Keywords:** endometriosis, natural killer cells (NK cells), NK receptors, endometrioma, ectopic endometrium, immune dysfunction, NK inhibition, NK suppression

## Abstract

Endometriosis is a chronic inflammatory disorder, characterized by the presence of endometrial cells outside the uterine cavity. An increasing number of studies correlate the immune system with endometriosis, particularly NK receptors (NKR), which have been suggested to play an essential role in the pathogenesis of the disease. This systematic review aims to enlighten the role of NKR in endometriosis. A literature search was performed independently by two reviewers, to identify studies assessing the role of NKR in endometriosis. In total, 18 studies were included. Endometriosis pathogenesis seems to be marked by the overexpression of NK inhibitor receptors (KIRS), namely, CD158a+, KIR2DL1, CD94/NKG2A, PD-1, NKB1, and EB6, and inhibiting ligands such as PD-L1, HLA-E, HLA-G, and HLA-I. Concurrently, there is a decrease in NK-activating receptors and natural cytotoxicity receptors (NCRs), such as NKp46, NKp30, and NKG2D. The immune shift from NK surveillance to NK suppression is also apparent in the greater relative number of ITIM domains compared with ITAM domains in NKRs. In conclusion, NK receptor activity seems to dictate the immunocompetency of women to clear endometriotic cells from the peritoneal cavity. Future research could explore NKRs as therapeutic targets, such as that which is now well established in cancer therapy through immunotherapy.

## 1. Introduction

Endometriosis is a chronic gynecological disorder characterized by the presence of endometriotic tissue outside the uterine cavity. It is a very incapacitating disease affecting 1 in 10 women during reproductive age [1], having a great impact on their quality of life. Although endometriosis was first noted by Carl von Rokitansky more than one hundred years ago, the pathogenesis of this condition is still not clear. Regarding pathogenesis, the theory of retrograde menstruation and implantation of ectopic endometrium is the most accepted [2]. Retrograde menstruation occurs in almost all women, but only 10% of them develop endometriosis [1]. Thus, the question arises as to what facilitates ectopic endometriotic cell implantation and growth. The hypothesis of immunological dysfunction has gained greater evidence in recent years. In the peritoneal cavity microenvironment, immune dysfunction enhances the capacity of ectopic endometrial cells to adhere and invade, leading to poor clearance of the ectopic endometrium and increased proliferation and angiogenesis [3]. NK cells have been suggested to play an important role in the pathogenesis of the disease by either tolerating or inhibiting the survival, implantation, and proliferation of endometrial cells [4]. An increasing number of studies [5,6,7,8,9,10,11,12,13,14,15,16,17,18,19,20] address the role of NK receptors (activating and inhibiting) in endometriosis pathogenesis (Figure 1). To the best of our knowledge, there are no systematic reviews concerning the involvement of NK cell receptors in endometriosis; therefore, this assessment is important for a better understanding of the topic.

## 2. Methods

This systematic review identified and analyzed human studies which assess the role of NK cell receptors in women suffering from endometriosis compared with healthy women, and was conducted according to the PRISMA guidelines (Preferred Reporting Items for Systematic Reviews and Meta-analyses) [21,22] (Figure 2) and registered in PROSPERO (CRD42020205783). 

### 2.1. Search Strategy

An extensive computerized literature search strategy was performed to retrieve the studies included in this review. The searches were performed in PubMed/Medline, EMBASE, Scopus, and Web of Science databases, using database-specific subject heading terms and all its variants in free-text words, according to the specificities of each database (Appendix A). To ensure literature saturation, we performed a supplementary search of grey literature and of the reference sections of the selected studies and reviews, to identify any additional relevant missing publications as well as conference papers not retrieved in the electronic search. No date or language limits were imposed on the search. 

### 2.2. Eligibility Criteria and Study Selection 

The study selection was performed independently by two individuals (J.L.R. and N.N.R.), who screened the titles and abstracts of all suitable articles from the queries according to the eligibility criteria (Appendix A). No restrictions to geographical location, language of publication, or year of publication were applied, and all non-primary studies were excluded.

Studies that did not meet eligibility criteria were excluded and the remaining articles were thoroughly analyzed and read in full and included or discarded according to the eligibility criteria. Any disagreement was resolved after discussion with a third reviewer (J.L.). All decisions, including reasons for exclusion and the number of selected articles in each step, were recorded and depicted in a flow chart, following the PRISMA 2020 guidelines.

EndNote version 20 was used to store, organize, and manage all the references arising from the literature search, including the management and removal of duplicates, and scanning of the titles/abstracts of all the records.

### 2.3. Data Extraction and Quality Assessment

All relevant data were extracted from each selected study by two independent reviewers and any disagreement was referred to a third reviewer (J.L.) to reach consensus. The following data were extracted: Identification of the study—title, authors, year of publication, journal title, country of origin, study design, number of participants, and recruitment procedure and duration.Participant characteristics—sample size, age, BMI, parity, rAFS classification of endometriosis [23], previous treatments, and preoperative painful symptoms scores—of subjects in endometriosis and control groups.Methodological features—sample characteristics (endometriotic tissue, peritoneal fluid, peripheral venous blood, and eutopic endometrium), methodology used for NK cell marker characterization (^51^Cr cytotoxicity assay, ELISA, flow cytometry, immunohistochemistry, in situ hybridization, RNA extraction, and quantitative real-time Western blotting) and NK cell markers.Outcomes—comparison of NK cell markers between the different samples studied.

The methodological quality of each individual study was assessed independently by two reviewers on 20 October 2022 (J.L.R. and N.N.R.), using the NHLBI quality assessment tool of case–control studies (https://www.nhlbi.nih.gov/health-topics/study-quality-assessment-tools). This scale evaluates 12 components of a study to determine its methodological quality. Each study evaluation criterion was assigned “Yes”, “No”, “NR” (not reported), “NA” (not applicable), or “Unc” (Unclear). Subsequently, studies were graded as “Poor”, “Fair”, or “Good”. The level of bias influences the evidence and results obtained in this systematic review. Conflicts regarding study quality were resolved with a third reviewer (J.L.)

### 2.4. Data Synthesis

Due to the high heterogenicity of the selected studies evaluating several NK cell receptors in different types of samples (peripheral blood, peritoneal fluid, and eutopic and ectopic endometrium), a quantitative meta-analysis was not feasible to perform. Data from the selected studies were narratively and qualitatively summarized.

## 3. Results

### 3.1. Study Selection

In total, 1193 studies were obtained. Upon the exclusion of 628 duplicates, titles and abstracts were screened, and 535 studies that did not meet eligibility criteria were excluded. There were 30 reports sought for retrieval, where an additional 8 studies were excluded because they were classified as conference abstracts/posters. The remaining 22 articles were thoroughly analyzed and read in full, leading to the exclusion of 4 more studies due to research theme incongruency. In total, 18 studies [5,6,7,8,9,10,11,12,13,14,15,16,17,18,19,20,24,25] were included in the final qualitative analysis, as per Figure 2. 

### 3.2. Study Quality Assessment and Risk of Bias

Quality assessment was conducted on the 18 selected studies as presented in Appendix A (following NHLBI Assessment Tool guidelines). Among these studies, 15 were deemed “good quality” [5,6,8,9,10,14,15,16,17,18,19,20,24,25]. This assessment was based on the sample size, consideration/adjustment to key confounding variables (demographic factors, menstrual cycle stage, hormonal therapy, and disease staging as per revised American Society for Reproductive Medicine (rASRM) classification) [23], and appropriateness of the methodology according to the research hypothesis. None of the studies included sample size justification or blinded evaluators of outcomes.

### 3.3. Heterogenicity of the Studies Assessed 

NK cell receptors (and their ligands) were evaluated in 18 case–control (observational) studies in the context of endometriosis, all using a human model. One of these was also classified as quasi-experimental [24], analyzing the impact of 17β-estradiol treatment on results. Heterogenicity was noted in the samples evaluated in each study, as presented in Table 1. Studies used a wide array of samples, including peritoneal fluid [6,7,8,9,10,11,13,14,16,17,18,19,24,25], endometriotic lesions/eutopic endometrium samples [12,15,18,19,24], and peripheral blood [7,8,9,10,18,20,24].

Additionally, each study employed different techniques to evaluate the NK cell receptor activity and expression in endometriosis. Flow cytometry [7,8,9,10,11,14,16,19,24,25] was most commonly used, followed by Western blotting [6,7,8,9,10,19,24], immunohistochemistry [12,15,18,24], polymerase chain reaction (PCR) [14,19,20], enzyme-linked immunosorbent assay (ELISA) [13,17,18], RNA in situ hybridization [12], and ^51^Cr cytotoxicity assays [5]. 

The study results are summarized in Table 1.

**Table 1 ijms-24-00331-t001:** Summary of findings and methods of selected studies regarding NK cell-activating/inhibiting receptors in endometriosis.

Samples	Design	n	Comparison	Methods	Markers	Results	Reference
PF	Case–control	11 I/II EDT; 22 III/IV EDT; 11 controls	NK cytotoxicity and NK inhibition receptors in early and late EDT	FC cytotoxicity assay ^51^Cr	FITC anti-CD45/PE-anti-CD14 γ_1_ FITC/γ_2a_ PE, FITC-anti-CD3/PE-anti-CD19, FITC-anti-CD3/PE-anti-CD56, FITC-anti-CD3/PE-anti-NKB1/PerCP-anti-CD56, FITC-anti-CD3/PE-anti-GL183/PerCP-anti-CD56, FITC-anti-CD3/PE-anti-EB6/PerCP-anti-CD56.	↓ peritoneal cytotoxicity against K562 EDT (I/II/III/IV) vs. controls; ↑ KIR expression (NKB1, EB6) in III/IV EDT vs. controls; ↑ KIR expression (NKB1, EB6) in III/IV EDT vs. I/II EDT	Wu et al., 2000 [5]
PF	Case–control	10 EDT; 10 controls	HLA-G inhibitory ligand expression in women with and without EDT	Western blotting	mAb aa 61-83 α1 domain of HLA-G	No statistical significance between groups	Hornung et al., 2001 [6]
PB, PF	Case–control	11 I/II EDT; 17 III/IV EDT; 6 controls	ICAM-1 and KIR expression in women with and without EDT	FC Western blotting	FITC-labeled anti-CD3 mAb, anti-CD4 mAb, PE-labeled anti-CD8 mAb, PE-labeled anti-CD19 mAb, FITC-labeled anti-CD16 mAb, FITC-labeled anti-CD14 mAb, PE-labeled anti-CD54 (ICAM-1) mAb, PE-labeled anti-CD158a, anti-CD158b, and CD94	↓ ICAM in PF macrophages EDT vs. controls; ↑ KIR2DL1+ NK among CD16+ NK in PB and PF of EDT vs. controls (more pronounced in III/IV EDT)	Maeda, Izumiya, Oguri et al., 2002 [7]
PB, PF	Case–control	12 I/II EDT; 30 III/IV; 40 controls	KIR2DL1+ NK cell expression in women with and without EDT	FC Western blotting	FITC-anti-CD16 mAb PE-labeled anti-CD158a mAb, anti-CD158b mAb, PE anti-CD94 mAb	↑ KIR2DL1+ NK cells in PB and PF of EDT vs. controls (more pronounced in III/IV EDT)	Maeda, Izumiya, Yamamoto et al., 2002 [8]
PB, PF	Case–control	18 I/II EDT; 70 III/IV EDT; 104 controls	CD158a+ cells (KIR subtype) expression in women with and without EDT	FC Western blotting	FITC-labeled anti-CD16 mAB, PE-labeled anti-CD158a and anti-CD158b mAbs, PE-labeled anti-CD94 mAbs	↑ CD158a+ cells in PF of EDT (I/II/III/IV and III/IV) vs. controls; ↑ CD158a+ cells in PB of EDT (I/II and III/IV)	Maeda et al., 2004 [9]
PB, PF	Case–control	6 I/II EDT; 18 III/IV; 25 controls	ITIM and ITAM KIR expression in women with and without EDT	FC Western blotting	FITC-labeled anti-CD56 mAb PE-labeled anti-CD158a mAb, anti-CD158b mAb, Anti-CD158a mAb, anti-CD158b mAb	ITIM-KIR > ITAM-KIR in PB EDT; ↑ CD158a+CD56+ NK cells in PB and PF EDT vs. controls	Matsuoka et al., 2005 [10]
Endometrium Peritoneal EDT	Case–control	15 EDT and 12 controls (IHC); 24 endometrium EDT and 14 peritoneal fluid EDT and 17 controls (RNA ISH)	HLA-G expression in eutopic and ectopic endometrium in women with and without EDT	IHC RNA ISH	mAb4H84 cDNA probe	HLA-G protein and gene transcripts found in >90% glandular peritoneal fluid EDT but not in stromal endometrial epithelium (controls and EDT)	Barrier et al., 2006 [12]
PF	Case–control	26 I/II EDT; 20 III/IV EDT; 24 controls	CD56+ cell expression, Fas antigen CD95 and early activation molecule CD69 in PF of women with and without EDT	FC	Anti-CD45FITC/CD14PE, IgG1FITC/IgG2aPE, anti-CD69FITC, anti-95FITC and anti-CD56PE	↓ CD56+ dim expression in I/II EDT vs. controls; ↑ CD95 in I/II EDT vs. controls; ↑ CD69+CD56+ in I/II EDT vs. controls; ↑↑ CD69+CD56+ in III/IV EDT vs. controls	Eidukaite et al., 2006 [11]
PF	Case–control	17 I/II EDT; 14 III/IV EDT; 27 controls	HLA-G expression in PF of women with and without EDT	ELISA	Anti-sHLA-G	No statistical significance between groups	Eidukaite and Tamosiunas, 2008 [13]
PB, PF, Endometrium Peritoneal EDT	Case–control	20 III/IV EDT; 13 controls	(HLA)-E receptor CD94/NKG2A expression in women with and without EDT	FC, RT-PCR	Anti-CD56 (clone C218), anti-NKG2A (clone Z199), PE-conjugated anti-NKG2A, PE-conjugated anti-NKG2C, fluorescein-conjugated anti-CD56, peridin chlorophyll protein-conjugated anti-CD3, anti-CD45, cytokeratin 20	↑ CD94/NKG2A cells in PF in EDT vs. controls; ↑ HLA-E mRNA in peritoneal EDT	Galandrini et al., 2008 [14]
Endometrium	Case–control	15 EDT; 15 controls	HLA-I and HLA-II expression in women with and without EDT	IHC	IgG2a (HLA-I), IgG1 (HLA-II)	↑ HLA-I and HLA-II expression in EDT endometrial stroma and glands vs. controls	Baka et al., 2011 [15]
PF	Case–control	3 I/II EDT; 18 III/IV EDT; 28 controls	Expression of NK Cell surface antigens (CD16 and CD56+ cells), NCRs (NKp46/40/30) and cytokine production (TNF-α, IFN-γ etc.) of PF NK cells in women with and without EDT	FC	Anti-CD45 PerCP-Cy5.5/anti-CD56 PE/anti-CD16 fluorescein isothiocyanate, anti-CD45 PerCP-Cy5.5/anti-CD56 FITC/anti-CD335 (NKp46) PE, anti-CD45 PerCP-Cy5.5/anti-CD56 FITC/anti-CD336 (NKp44) PE, and anti-CD45 PerCP-Cy5.5/anti-CD56 FITC/anti-CD337 (NKp30) PE	↓ NKp46+ NK cells in III/IV EDT vs. controls; ↓ CD56dim/NKp46+ cells in III/IV EDT vs. controls; ↑ TNF-α producing NK cells in III/IV EDT vs. controls; ↑ IFN-γ producing NK cells in III/IV EDT vs. controls	Funamizu et al., 2014 [16]
PF	Case–control	121 EDT; 81 controls	Levels of soluble NKG2D ligands (MICA, MICB and ULBP-2) in women with and without EDT	ELISA	ELISA (R&D Systems, Inc., Minneapolis, MN, USA)	↑ MICA in EDT vs. controls; ↑ MICB in EDT vs. controls; ↑ MICA, MICB, ULBP-2 in deep infiltrating EDT	González-Foruria et al., 2015 [17]
Serum, PF, Endometrium EDT	Cross-sectional observational/Case–control	60 I/II EDT; 83 III/IV EDT; 77 controls (ELISA) 26 EDT; 22 controls (IHC)	Soluble HLA-G expression in endometrium, EDT, PF, and serum in women with and without EDT	ELISA IHC	mAb 4H84 MEM-G/9 mouse mAb	↑ sHLA-G in serum but not PF of III/IV EDT vs. controls; ↑ HLA-G protein expression in EDT but not eutopic endometrium	Rached et al., 2019 [18]
Endometrium EDT PB	Case–control Quasi-experimental	15 EDT; 15 controls	PD-1/PDL-1 expression in women with and without EDT, post and prior estrogen and cytokine treatment	IHC FC Western blotting	Rabbit anti-human PD-1, CD4, CD8, PD-L1. anti-GAPDH CD279 (PD-1)-PE, CD274 (B7-H1)-PE, CD8-FITC, and CD4-PerCP-Cyanine5.5	↑ PD-1/PDL-1 and PD-1 in eutopic and ectopic endometrium EDT vs. controls; ↑ PD-1/PDL-1 and PD-1 in PB EDT vs. controls; ↑ PD-1/PDL-1 and PD-1 in eutopic endometrium EDT post 17β-estradiol treatment vs. controls	Wu et al., 2019 [24]
Endometrium EDT PF	Case–control	20 EDT; 13 controls	NKG2D expression and its ligands in women with and without EDT	FC RT-PCR Western blotting	FITC-conjugated mouse, anti-human CD16 mAb, PE-Cy5-conjugated mouse, anti-human CD56 mAb PE-mouse, anti-human NKp30 mAb, PE-mouse, anti-human NKp44 mAb, PE-mouse, anti-human NKp46 mAb, PE-mouse, and anti-human NKG2D mAb	↓ NKp30 CD56+ in PF in EDT; ↑ NKp46 CD16+ in PF in EDT; ↓ NKG2D CD56+ in PF in EDT; ↓ ULBP-2 in eutopic endometrium in EDT vs. controls; ↓ ULBP-3 in ectopic endometrium in EDT vs. controls and eutopic EDT	Xu, 2019 [19]
PB	Case–control	147 EDT; 117 controls	HLA-C and KIR polymorphism combinations in women with and without EDT	PCR	Commercial kits (HLAssure SE kit, Accutype Software, SSO typing kit)	↑ HLA-C*03:03:01 occurrence in EDT vs. controls; ↑ KIR centromeric A/A haplotypes in EDT vs. controls; ↓ KIR2DS2-positive individuals in EDT vs. controls	Chou et al., 2020 [20]
PF	Case-control	32 EDT; 30 controls	NKp46 co-expression patterns in women with and without EDT	FC	anti-TNF-α-BV421, anti-IFN-γ-PE- Cy7, anti-IL-4-PerCP-Cy5, anti-IL-10- APC, anti-TGF-β-PE	↓ CD56+/NKp46+ in PF in EDT; ↓ NKG2C+/NKp46+ in PF in EDT; ↓ CD16+/NKp46+ in PF in EDT; ↑ NKG2D+/NKp46+ in PF in EDT No significant co-expression of inhibitory receptors (CD158a and NKG2A) and NKp46 ↑ IFN-γ-producing NK cells in PF in EDT	Saeki et al., 2022 [25]

aa, amino acids; EDT, endometriosis; FC, flow cytometry; IHC, immunohistochemistry; ISH, in situ hybridization; ITAM, tyrosine-based activation motifs; ITIM, tyrosine-based inhibitory motifs; KIR, natural killer inhibitor receptor; mAb, monoclonal antibody; PB, peripheral venous blood; PF, peritoneal fluid. Scoring and staging of EDT (I/II/III/IV) as per American Fertility Society Guidelines [23].

### 3.4. NK Cell Receptors in Peritoneal Fluid

Fourteen studies evaluated the activity and expression of NK cell receptors in the peritoneal fluid of endometriosis subjects [5,6,7,8,9,10,11,13,14,16,17,18,19,25]. The role of immune dysfunction in the progression of endometriosis was highlighted by Maeda and colleagues [7,8,9], who found, by using flow cytometry, that the levels of NK cell inhibitory receptors such as CD158a+ and KIR2DL1+ were significantly higher in the peritoneal fluid of endometriosis subjects than controls. This increase was even more pronounced in stage III/IV endometriosis (KIR2DL1+ 24.2% ± 16.1% vs. 12.3% ± 6.9%, *p* = 0.007 [8], 17.5% ± 8.7% vs. 11.7% ± 5.5%, *p* = 0.008 [7], CD158a+ 17.0% ± 12.3% vs. 12.3% ± 7.1%, *p* = 0.017 [9]) Matsuoka et al. [10] corroborated these findings, also reporting an increased percentage of CD158a+ in CD56+ NK cells in the peritoneal fluid of the endometriosis group (13.9% ± 7.5% vs. 7.6% ± 3.0%, *p* = 0.028). Galandrini et al. [14] conducted a similar study using cytofluorometric analysis and reported an increased frequency of human leukocyte antigen–E inhibitory receptor CD94/NKG2A-expressing peritoneal NK cells in subjects with endometriosis (75.9% ± 8.2% vs. 52.1% ± 16.3%, *p* < 0.0001). Wu et al. [5] also explored NK inhibitor receptor (KIR) expression in endometriosis with flow cytometry, noting a significant increase in the mean fluorescence intensity of receptors NKB1 (183 ± 56 vs. 97 ± 45, *p* < 0.001) and EB6 (82 ± 45 vs. 53 ± 14, *p* = 0.021) in the peritoneal fluid of subjects with severe (III/IV) endometriosis compared with the controls. The increased inhibition of NK cell activity at the peritoneal level in endometriosis was also explored through NK cell inhibitory ligands in three studies [6,13,18], which analyzed HLA-G expression in peritoneal fluid using different methods (ELISA, Western blotting, and immunohistochemistry). All the studies reached the same result: there were no significant statistical differences in HLA-G in the peritoneal fluid of endometriosis subjects compared with the controls. These results were different in other samples ((serum and endometrium) [12,18]), as expressed in the following sections of this report.

NK cell-activating receptors were also investigated in the studies selected. Using flow cytometry, Xu [19] reported a decrease in NKp30 (*p* = 0.006) and NKG2D (*p* = 0.01) levels in CD56+ NK cells in the peritoneal fluid of endometriosis subjects. In this same study, however, Xu noted a significant increase in activating receptor NKp46 in CD16+ NK cells (*p* = 0.04) 19. This finding is in contrast to that of Funamizu et al. [16], who reported a decrease in peritoneal activating receptor NKp46 in advanced (III/IV) endometriosis: 22.84% (11.58–46.34) vs. 30.20% (18.72–51.20), *p* < 0.05. Furthermore, these authors noted a decrease in peritoneal CD56dim/NKp46 cells, 16.67% (8.95–33.09) vs. 26.12% (14.60–41.14, *p* < 0.05), as well as an increase in TNF-α (33.87% (25.21–60.73) vs. 25.42% (17.56–30.43), *p* < 0.05) and IFN-γ (50.04% (26.98–71.33) vs. 19.41% (6.01–38.37), *p* < 0.05) producing NK cells [16]. This decrease in peritoneal cytotoxic cells (CD56dim/NKp46) in endometriosis is complemented by the findings of Wu et al. [5], whose ^51^Cr release assay showed a decrease in cytotoxicity against K562 in the peritoneal fluid of subjects with severe (III/IV) (LU_20_ 4.6 ± 3.6 vs. 8.0 ± 4.0 *p* = 0.011) and early (I/II) (LU_20_ 4.7 ± 2.7 vs. 8.0 ± 4.0 *p* = 0.028) endometriosis. The decreased activation of NK cell activity at the peritoneal level in endometriosis was also explored through ligands. González-Foruria et al. [17] noted an increase in soluble forms or the shedding of NK-activating receptor ligands (namely, MICA (*p* = 0.015), MICB (*p* = 0.003), and ULBP-2 (0.045)) in the peritoneal fluid of subjects with deep infiltrating endometriosis relative to controls. Further elucidating the role of NK cell-activating receptors, Saeki et al. [25] noted a decrease in co-expression of CD56+/NKp46+ and CD56dim/NKp46+ cells (*p* < 0.05) with a negatively correlated increase in IFN-γ-producing cells (*p* < 0.01) in subjects with endometriosis. Contrastingly, they also observed an increase in the co-expression of activating receptor NKG2D+/NKp46+ cells in endometriosis subjects (*p* < 0.05). 

NK cell activity may also be suppressed through the loss of NK cells (apoptosis). Eidukaite et al. [11] implicated the Fas–FasL mechanism in the pathophysiology of early endometriosis by reporting a cytometric decrease in CD56dim NK cells (24.0% ± 4.9% vs. 36.3% ± 15.9%, *p* < 0.02), and an increase in CD95 (Fas antigen that promotes NK apoptosis) (66.6% ± 10.6% vs. 42.4% ± 10.1%, *p* < 0.01) in the peritoneal fluid of subjects with early endometriosis. They also reported an increase in NK cells containing the Fas early activation molecule (i.e., CD69+ CD56+ NK cells) in the peritoneal fluid of subjects with early (66.8 ± 16.5% vs. 43.3 ± 13.7%, *p* < 0.01) and advanced (74.2% ± 7.9% vs. 43.3% ± 13.7%, *p* < 0.01) endometriosis [11].

### 3.5. NK Cell Receptors in Endometrium and Endometriotic Lesions

Endometrium samples and endometriotic lesions were assessed for the expression of a variety of NK receptors/ligands in the studies selected, to ascertain which features promoted immune evasion and ectopic proliferation/implantation. Two studies [12,18] explored HLA-G (inhibitory NK cell ligand) protein expression in the endometrium and endometriotic lesion samples via immunohistochemistry. As per Barrier et al. [12], HLA-G protein was found in the glandular epithelia of 14 out of 15 peritoneal endometriotic lesion samples (93.3%) (none in stroma), whereas none was found in eutopic endometrium of both control and endometriosis samples. Similarly, Rached et al. [18] found a significant increase in the HLA-G protein expression in endometriotic lesions of advanced-stage endometriosis compared with the eutopic endometrium (both control and endometriosis) (*p* = 0.018). Barrier et al. [12] went further to analyze HLA-G gene transcripts in endometrial samples using RNA in situ hybridization. With this technique, they found that the glandular epithelia of 13 out of 14 peritoneal lesions (92.8%) contained the HLA-G gene transcript, but this was not present in the eutopic endometrium (control and endometriosis samples). The phenomenon of NK cell inhibition was also explored through the expression of programmed-death 1 (PD-1) and its ligand (PD-L1) in endometriosis using immunohistochemistry and Western blotting [24]. Immunohistochemistry results showed an increase in PD-1 and PD-L1 expression in the eutopic and ectopic endometria of endometriosis samples compared with the controls (*p* < 0.05 and *p* < 0.01, respectively) [24]. PD-L1 of the eutopic endometrium of endometriosis samples was upregulated by 17β-estradiol treatment (*p* < 0.01). Regarding Western blot results, ectopic endometriotic lesions showed a significant increase in PD-1 and PD-L1 expression compared with the controls and the eutopic endometrium from endometriosis samples (*p* < 0.01); eutopic endometrium from endometriosis also differed significantly from the controls, exhibiting greater PD-L1 expression (*p* < 0.01) [24]. Additionally, a significant increase in CD4+/CD8+ T cell infiltrate occurrence and size was reported in endometriosis samples (eutopic and ectopic) compared with controls (*p* < 0.01) [24]. In contrast, in the context of activation, NK cell-activating receptor NKG2D ligands were assessed via real-time PCR and Western blotting. Although Xu [19] did not find any statistically significant differences in ligand mRNA expression, changes at a translational level were observed, namely, a significant decrease in ULBP-2 protein expression in the eutopic endometrium of endometriosis samples (relative to the eutopic endometrium of control samples and ectopic lesions (*p* < 0.05)), and a decrease in ULBP-3 protein expression in the ectopic endometrium samples compared with the eutopic endometrium (control and endometriosis) (*p* < 0.05). Factors of immune evasion were also measured [15]. Using immunohistochemistry on endometrial samples obtained via curettage from endometriosis samples and controls, Baka et al. [15] noted a significant increase in HLA-I and HLA-II expression in both the glandular (HLA-I 87.5% vs. 56.3%, *p* < 0.02, HLA-II 46.7% vs. 20.0%, *p* < 0.04) and stromal (HLAI 100% vs. 87.5%, *p* < 0.02, HLAII 66.7% vs. 40.0%, *p* < 0.007) epithelia of endometriosis samples compared with the controls.

### 3.6. NK Cell Receptors in Peripheral Blood and Serum

Several studies [7,8,9,10,18,20,24] have explored NK cell receptor expression in the peripheral venous blood of endometriosis patients. Chou et al. [20] investigated the impact of HLA-C (KIR ligand) and KIR genotype/polymorphisms on the risk of endometriosis in a Han Chinese cohort using a DNA-based method of PCR amplification. They showed that a genotype combination HLA-C*03:03:01 was associated with an increased risk of endometriosis (OR = 2.811, *p* = 0.0473), while the presence of KIR2DS2 acted as a protective factor against endometriosis with an odds ratio of 0.5577 (*p* = 0.0394). Moreover, the number of KIR centromeric A/A haplotypes was significantly increased in the endometriosis group compared with the control group (OR = 1.793, *p* = 0.0394) [20]. Maeda et al. [9] studied the expression of NK inhibitory receptor CD158a+ in the peripheral blood of endometriosis subjects using flow cytometry and observed a statistically significant increase of 3.8% (17.5% ± 9.9% vs. 13.8% ± 7.7%, *p* = 0.009) in the endometriosis group compared with the control group. Matsuoka et al. [10] observed similar flow cytometry results, reporting a significant increase in CD158+ CD56+NK cells in the endometriosis group compared with the control group (18.8% ± 8.2% vs. 13.4% ± 5.8%, *p* = 0.021). Additionally, the authors also noted a trend whereby KIRs possessing ITIM domains were predominant over KIRs possessing ITAM domains (65.6% ± 32.1% vs. 3.9% ± 2.7%) in the endometriosis group’s peripheral blood samples [10]. Other NK cell inhibitory receptors such as KIR2DL1 + NK cells were also explored in the studies selected [8]. Similarly to CD158a+, endometriosis KIR2DL1 + NK cell levels were significantly higher than those of the control groups (17.4% ± 8.4% vs. 11.7% ± 5.5%, *p* = 0.011) [8]. Another study by Maeda et al. [7] corroborated this, demonstrating higher levels of KIR2DL1 + NK cells among CD16 + NK cells in the endometriosis group compared with the control group; however, the results were only statistically significant in advanced endometriosis (24.3% ± 15.6% vs. 11.6% ± 6.8%, *p* = 0.008). Soluble levels of the NK cell inhibitory ligand HLA-G were also investigated by Rached et al. [18] in serum, using ELISA, reporting significantly higher levels of sHLA-G in sera in the endometriosis group compared with the controls (9.3 U/mL (1.2–192.6) vs. 6.1 U/mL (1.0–36.9), *p* = 0.013). Lastly, natural cytotoxicity receptors (NCRs) and NKG2D (NK cell-activating receptors) were studied in the peripheral blood of endometriosis patients by Wu et al. [24]. Unlike the significant differences observed in the peritoneal fluid mentioned above, peripheral blood yielded no statistically significant differences regarding NCRs or NKG2D expression between endometriosis and control groups [24].

## 4. Discussion

Endometriosis is a multi-factorial disease, characterized by genetic propensity, environmental insult, and, as highlighted in this review, immune dysfunction [26]. Following Sampson’s theory of retrograde menstruation, tissue fragments and cells must not only escape the uterine cavity, but also evade apoptosis, adhere to ectopic tissue, generate a local vascular supply, and evade immune surveillance systems to develop into endometriotic lesions [2,26]. NK cell suppression via receptor-mediated dysfunction plays a crucial part in evading immunosurveillance and apoptosis, enabling the proliferation and adhesion of ectopic endometrial cells [27]. Secondary to inflammation associated with endometriosis, the NK cell-mediated release of cytokines and pro-angiogenic factors such as TNF-α and IFN-γ creates an optimal microenvironment for the proliferation of ectopic endometriotic lesions [16]. The studies explored in this review [5,6,7,8,9,10,11,12,13,14,15,16,17,18,19,20,24] corroborate this notion, detailing a pathological local environment of endometriosis whereby the aberrant expression/activity of NK cell receptors results in less cytotoxicity (via the increased inhibition, decreased activation, and apoptosis of NK cells themselves) and contribute to a pro-inflammatory state which promotes proliferation and angiogenesis. NK cell suppression has been implicated in the pathogenesis of endometriosis, given its role in immunocompetency and the clearance of atypic debris. 

The increased apoptosis of NK cell populations (through aberrant expression of NK cell death receptors) results in increased immune evasion, leaving dysregulated cellular mechanisms and proliferation unconstrained. This is observed in endometriosis, as evidenced by Eidukaite et al. [11] through their flow cytometry results: endometriosis samples do not only contain decreased levels of CD56+ cytotoxic NK cells in peritoneal fluid, but also exhibit an increased presence of Fas antigen (CD95) and increased levels of early-activation molecules (i.e., CD69) in the initial stages of endometriosis (I/II). This suggests that, in the early stages of endometriosis, the Fas–FasL mechanism of induced cell death is overexpressed, leading to a decrease in NK cell populations, and thus, the increased immune evasion of ectopic endometriotic cells [11]. 

NK cell inhibitor receptors have also been implicated in immune dysfunction associated with endometriosis. Inhibitor receptor overexpression appears to be a feature of endometriosis, evidenced by several of the studies explored. Matsuoka et al. [10] observed that ITIM (inhibition) receptor domains were more prevalent than ITAM (activation) receptor domains in NK cells found in the peripheral blood of endometriosis subjects. This suggests that the immune balance of activation/inhibition is shifted in endometriosis, favoring NK cell suppression, and thus promoting ectopic endometrial cell survival. This study, however, had the limitation of only having analyzed ITIM/ITAM domains in the peripheral blood of subjects. Most ectopic implantation sites are in the peritoneal cavity; therefore, the peritoneal fluid analysis of ITAM and ITIM domain levels of local NK cells may be more accurate in depicting endometriotic disease profiles than the peripheral blood. Matsuoka et al. [10] also identified a significantly greater expression of NK inhibitor receptor CD158a in CD16+ NK cells in the peripheral blood and peritoneal fluid of endometriosis subjects compared with controls. This finding is supported by Maeda et al. [7,8], who also reported an increase in inhibitor receptor CD158a+ NK cells in the peritoneal fluid and peripheral blood of endometriosis subjects [9]. Unlike Matsuoka et al. [10], however, Maeda et al. [7,8] also found a significant overexpression of NK inhibitor receptor KIR2DL1; the results yielded by Matsuoka et al. [10] did not show significant results for KIR2DL1 levels. This could be due to variability in study populations—it may be that subjects studied by Maeda et al. [7,8] had a greater relative disease severity than those studied by Matsuoka et al. [10], because it is recognized that KIRD2DL1 overexpression appears to be progressive. Stage III/IV endometriosis exhibits significantly greater KIR2DL1+ NK levels than stage I/II samples. As for other inhibitor receptors, Galandrini et al. [14] reported a significant increase in CD94/NKG2A receptors in the peritoneal fluid of endometriosis subjects compared with controls. All these results support the hypothesis of NK cell inhibition being one of the underlying factors promoting the immune evasion of ectopic endometrial cells. Maeda et al. [8] also propose that the level of inhibition is potentially correlated with disease severity. A limitation of these studies, however, is that although they report greater KIR expression, the methodologies did not explore whether this indeed reduces NK cell activity. A complementary cytotoxicity assay would be appropriate to confirm the hypothesis. Wu et al. [5] bridged this knowledge gap by conducting a ^51^Cr cytotoxicity assay against K562 cells. They found that in all stages of endometriosis, there was significantly less peritoneal cytotoxicity against K562 cells than in controls [5]. Additionally, they observed a significant increase in KIR (NKB1, EB6) expression, which appeared to be correlated with disease severity (III/IV was significantly greater than I/II endometriosis) [5]. These findings together support the notion that NK cell activity is suppressed via KIR overexpression in endometriosis. Notably, in these studies, it is unclear whether KIR overexpression and reduced cytotoxicity were the cause or the result of endometriosis. To elucidate this, the influence of the genetic background on KIR expression/activity could be considered. As reported by Chou et al. [20], specific gene polymorphisms which affect inhibitor receptors such as HLA-C*03:03:01 and KIR centromeric A/A haplotypes occurred more often in subjects with endometriosis than controls. This finding suggests that altered KIR expression and immune dysfunction in endometriosis is associated with individual genetic backgrounds; specific genetic profiles seemed to be risk factors, associated with the higher occurrence of endometriosis. This defends the notion that KIR overexpression has a causal relationship with endometriosis, rather than occurring because of the disease. It is pertinent to consider, however, that this study was conducted in a Han Chinese cohort [20]. Whether these genetic findings would also apply to the general population is uncertain. Studies with more diverse samples are necessary to ascertain whether the extrapolation of findings to the general population would be accurate.

Ligands that bind to NK inhibitor receptors are also upregulated in endometriosis. This is evidenced, for instance, by the finding of significantly higher levels of HLA-E mRNA (which binds to the C-type lectin protein HLAI-specific inhibitory NK receptor CD94/NKG2A) in the peritoneal fluid of endometriosis subjects compared with controls [14]. Similar results have been reported on ligands which bind to the other family of HLA-I-specific inhibitory NK receptors or killer immunoglobulin-like receptors. HLA-I is an established inhibitory ligand that binds to KIRs on NK cells to prevent autoimmune responses; cells missing an HLA-I complex are targeted by NK cells for cell death following the “missing-self hypothesis” [28]. As per Baka et al. [15], another mechanism of endometrial immune evasion is the overexpression of HLA-I and HLA-II in both the glandular and stromal epithelia of endometrial tissue. The higher levels of HLA-I and HLA-II favor the suppression of cellular immunity in the peritoneal cavity, providing optimal conditions for the establishment of ectopic lesions. Nevertheless, this experiment is not without fault; Baka et al. only investigated samples of eutopic endometrium (found within the uterine cavity); therefore, it is unclear whether these expression trends are maintained in ectopic sites [15].

HLA-G, another NK receptor inhibitory ligand, has a stress-inducible gene; therefore, it is commonly found in tumors or in states of cellular stress, facilitating immune evasion and atypic cellular survival [29]. The selected studies explored this molecule extensively, mainly noting the absence of significant expression in the peritoneal fluid of endometriosis subjects [6,13]. Rached et al. [18] contradicted these findings by noting an increase in soluble HLA-G levels in the peritoneal fluid of endometriosis subjects in the menstrual phase compared with the secretory phase, although this increase was not statistically significant. These results align with Sampson’s theory [2] of retrograde menstruation, whereby endometrial cells escape to the peritoneal cavity during menstruation via patent fallopian tubes. Rached et al. [18] also observed significantly increased levels of soluble HLA-G in the serum of III/IV endometriosis subjects. Additionally, a significant increase in HLA-G protein and gene expression in ectopic peritoneal endometrial lesions’ glandular epithelium, but not the eutopic endometrium, was observed [12,18]. According to these results, it is suggested that ectopic endometrial lesions, upon encountering a new local immune environment, express HLA-G to promote cell survival and immune evasion via NK cell inhibition: this expression appears to be localized to the tissue fragments only. Furthermore, the absence of HLA-G expression in peritoneal fluid highlights that immune dysfunction associated with endometriosis derives from both NK inhibitor receptor and ligand overexpression.

NK immune checkpoint/inhibitory receptor programmed death-1 (PD-1) and its ligand, PD-L1, have also been investigated [30]. As per Wu et al. [24], increased proportions of PD-1- and PD-L1-positive cells were found in the endometriosis endometrium and blood samples compared with controls. The PD-1/PD-L1 axis has been thoroughly investigated in the oncological setting, and these molecules seem to be upregulated in a variety of cancer cells (ovarian/gastric carcinoma/leukemia) [31,32,33]. Thus, it has been postulated that PD-1 and PDL-1 serve as immunoinhibitory mechanisms contributing to tumor evasion from host defensive responses. These results suggest that this type of immune dysfunction also occurs in the pathogenesis of endometriosis. Further supporting this notion is the fact that PD-1/PDL-1 and PD-1 levels are upregulated in the eutopic endometrium of endometriosis samples following 17β-estradiol treatment. These results align with the established knowledge of endometriosis being an estrogen-dominant condition, whereby estrogen exposure promotes established ectopic endometrial lesion growth and exacerbates associated pain and inflammation [34]. Additionally, Wu et al. [24] noted a significant increase in the occurrence and size of CD4+/CD8+ T cell infiltrates in endometriosis samples. This observation is congruent with the inflammatory landscape of endometriosis, where the increased T cell-mediated release of mediators results in higher levels of circulating cytokines. Funamizu et al. [16] also explored the endometriotic cytokine environment, observing an increase in TNF-α- and IFN-γ-producing NK cells in III/IV endometriosis compared with controls. Saeki et al. [25] corroborates these findings, having also observed an increase in IFN-γ-producing NK cells in endometriosis. This aligns with the pathogenic setting of endometriosis as cytokines promoting an inflammatory state, optimal for angiogenesis and tissue invasion required for ectopic lesion establishment and survival.

The suppression of NK cell-activating receptors has also been implicated in immune dysfunction associated with endometriosis. Funamizu et al. [16] investigated this phenomenon, observing differences in the expression of several NCRs such as NKp46, NKp40, and NKp30. Of these, only NKp46 receptors exhibited a significant decrease in III/IV endometriosis compared with the controls [16]. Nevertheless, this decrease in receptor expression suggests that NK cell cytotoxicity suppression in endometriosis is also achieved by decreased activation. Xu [19] examined NKG2D receptor levels and respective ligands (MICA, MICB, ULBP-2, and ULBP-3), finding a significant decrease in NKG2D in CD56+ NK cells in the peritoneal fluid of endometriosis subjects, decreased expression of ULBP-2 in the eutopic endometrium of endometriosis subjects, and decreased ULBP-3 expression in ectopic endometrial lesions. Furthermore, González-Foruria et al. [17] explored NKG2D ligands. Unlike Xu [19], these authors found a significant increase in MICA, MICB, and ULBP-2 in the peritoneal fluid of subjects with deep infiltrating endometriosis compared with controls, and proposed that this increase was due to the shedding of ligands from endometrial cells onto the peritoneal space, resulting in the lower expression of these ligands on ectopic endometrial cell surfaces [17]. In turn, this promotes immune evasion and decreased NK cell cytotoxicity (less activation/triggering of NK cells by the ectopic lesions). It is possible that the results may not be fully representative of the ligand environment in endometriosis, whereby some allelic variants of MICA are not shed into the peritoneal space, and thus compensate for the absence of other allelic forms. This, although unlikely, cannot be confirmed due to the lack of literature on NKG2D ligands in endometriosis. Chou et al.’s [20] methodology may also be used to ascertain whether this apparent decrease in NK-activating receptors is the cause or the result of endometriosis. As per the study’s observations, there were fewer KIR2DS2-positive individuals in the endometriosis cohort compared with the controls [20]. KIR2DS2 (a gene encoding for an NK-activating receptor) appears to be a protective factor against endometriosis, suggesting that the inherent presence of fewer activating receptors creates an optimal environment for the development of endometriosis. However, as mentioned before, given the specificity of the cohort chosen in this study [20], it is unclear whether these results can be extrapolated to the general population. Saeki et al. [25] observed an increase in co-expression of NKp46+ and activating receptor NKG2D+. This prompts the hypothesis that aberrant or excess expression of NK cell-activating receptors may also be involved in the immune dysfunction of endometriosis. When chronically stimulated, NKG2D may exert inhibitory effects, decreasing missing-self signaling, as observed in murine models [25,35].

## 5. Conclusions

The overall objective of this systematic review was to assess the role of NK receptors (activating and inhibiting) in endometriosis pathogenesis, as depicted in the current literature. This was achieved by analyzing a total of 18 studies [5,6,7,8,9,10,11,12,13,14,15,16,17,18,19,20,24,25]. As per their results, endometriosis pathogenesis is marked by the overexpression of NK inhibitor receptors (KIRS), namely, CD158a+, KIR2DL1, CD94/NKG2A, PD-1, NKB1, and EB6, and inhibiting ligands such as PD-L1, HLA-E, HLA-G, and HLA-I in ectopic endometrial lesions. Concurrently, there is also a decrease in NK-activating receptors and NCRs such as NKp46, NKp30, and NKG2D, and a decreased expression (or increased shedding into the peritoneal space) of NK-activating ligands such as MICA, MICB, ULBP-2, and ULBP-3. Additionally, aberrant/excess expression of NKG2D+ has also been suggested to decrease immune response to ectopic endometrial cells. The notion of an immune shift from NK surveillance to NK suppression is also apparent in the greater relative number of ITIM domains compared with ITAM domains in NK receptors found in endometriosis subjects. Further exacerbating conditions for immune evasion and ectopic proliferation are NK cell death via the Fas–FasL mechanism in early endometriosis, decreased NK cell cytotoxicity, and increased NK cell-mediated cytokine release of TNF-α and IFN-γ, a state which promotes inflammation and angiogenesis required for ectopic endometrial cell lesion survival. Immune dysfunction associated with endometriosis also appears to have a genetic component—certain genotypes, namely, those which cause NK receptor dysfunction, have been associated with a greater risk of endometriosis. Overall, NK receptor activity dictates the immunocompetency of women to clear ectopic endometrial lesions from the peritoneal cavity. The suppression of NK cell activity (either by over-inhibition or decreased activation) creates optimal conditions for the development of endometriosis because it promotes lesion immune evasion, proliferation, and survival. If it is proven that the immune escape of endometriosis cells, allowing their implantation and growth, is related to the function of NK cell receptors, this may open the door to the application of immunotherapy in the treatment of severe endometriosis, as is already the case today with NK inhibitor receptor blockers in cancer treatment [36]. Further research and understanding on NK cell receptor involvement in endometriosis potentiates the development of novel therapeutic strategies, bringing about improvements in patients’ quality of life (clinical significance) and a novel approach towards endometriosis-associated infertility. Future research could explore NK inhibitor receptors as therapeutic targets, counteracting their overexpression in endometriosis with a local antagonist.

## Figures and Tables

**Figure 1 ijms-24-00331-f001:**
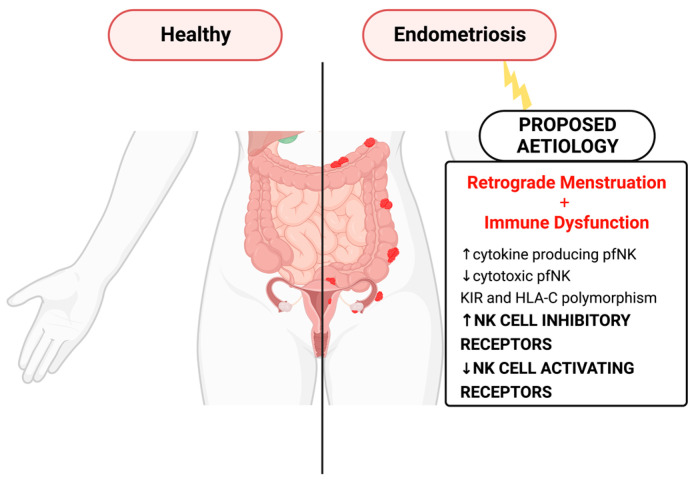
Diagrammatic representation of endometriosis and proposed etiology.

**Figure 2 ijms-24-00331-f002:**
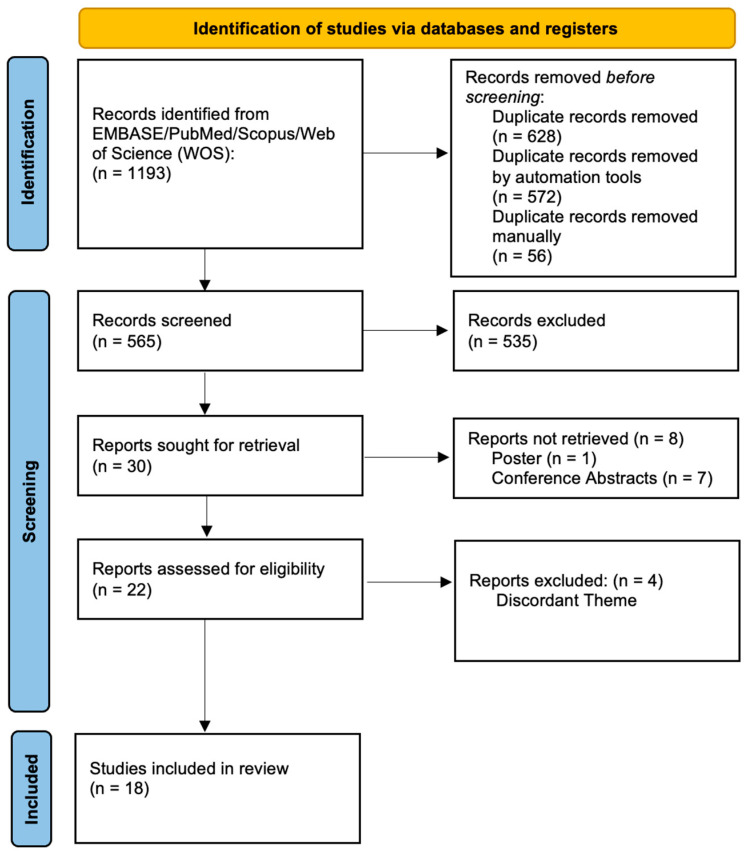
PRISMA flowchart [22].

## Data Availability

Not applicable.

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
