# Peer review of "Natural Killer Cell Receptors and Endometriosis: A Systematic Review"

_ijms, 2022, doi:10.3390/ijms24010331_

Round 1

Reviewer 1 Report

The detailed mechanism underlying endometriosis development remains unclear; few reports have suggested the involvement of immune and genetic factors. This systematic review aims to enlighten the role of NKR in endometriosis. A literature search was performed independently by two reviewers, to identify studies assessing the role of NKR in endometriosis using the PRISMA method. I greatly appreciate the effort put into using this method of publishing. However, I would like to ask you to re-analyse the literature, because the number of cited papers is surprisingly small for a review paper. The latest, very interesting items are especially lacking, e.g.
1. Co-expression of activating and inhibitory receptors on peritoneal fluid NK cells in women with endometriosis. Saeki S, Fukui A, Mai C, Takeyama R, Yamaya A, Shibahara H. J Reprod Immunol. 2022 Nov 21;155:103765. doi: 10.1016/j.jri.2022.103765.
2. Hypoxia-hindered methylation of PTGIS in endometrial stromal cells accelerates endometriosis progression by inducing CD16- NK-cell differentiation. Peng H, Weng L, Lei S, Hou S, Yang S, Li M, Zhao D. Exp Mol Med. 2022 Jul;54(7):890-905. doi: 10.1038/s12276-022-00793-1. Epub 2022 Jul 4. PMID: 35781537 Free PMC article.
3. The Promises of Natural Killer Cell Therapy in Endometriosis. Hoogstad-van Evert J, Paap R, Nap A, van der Molen R. Int J Mol Sci. 2022 May 16;23(10):5539. doi: 10.3390/ijms23105539.
4. Eutopic endometrial immune profile of infertility-patients with and without endometriosis. Freitag N, Baston-Buest DM, Kruessel JS, Markert UR, Fehm TN, Bielfeld AP. J Reprod Immunol. 2022 Mar;150:103489. doi: 10.1016/j.jri.2022.103489.
In these publications we find answers to a number of doubts and questions posed by you in your article. In conclusion, the work is quite interesting and important for the immunogenetics of endometriosis, but needs updating.

Author Response

Reply to Reviewer #1

The authors appreciate the review and comments made, which are extremely pertinent and useful for the work. In this review the authors wanted to do a compilation of existing studies, in humans, that specifically addressed Natural Killer cell receptors in women with endometriosis, i.e. at the level of ectopic endometrial cells. Effectively the number of published studies focusing on this specific topic are few, as the reviewer very well noted. The authors consider this theme of special importance. This topic had special development in the area of oncology, realizing that an important mechanism of immune escape from cancer cells to the immune system was verified by changing the receptors of NK cells, making them less capable of response. Such a mechanism can also be used by endometrial cells to escape the immune system and implant themselves outside the uterine cavity. The authors of this review carry out research in this area and hope to contribute, very soon, to the scientific community with their own results in this very interesting area of immunology.

The authors finished the selection of articles in July 2022, reserving the last months for the preparation of the manuscript. Following the reviewer's recommendation, the authors performed a new review using the rigorous criteria identified in the methodology and focusing on the last fewest months, finding 1 very important article that meets our selection criteria, so we have already included it in actual revision (“Saeki S, Fukui A, Mai C, Takeyama R, Yamaya A, Shibahara H. Co-expression of activating and inhibitory receptors on peritoneal fluid NK cells in women with endometriosis. J Immunol Reprod. 2022 Nov 21; 155:103765"). This article had been suggested by the reviewer, which we greatly appreciate.

The suggested article "Hoogstad-van Evert J, Paap R, Nap A, van der Molen R. The Promises of Natural Killer Cell Therapy in Endometriosis. Int J Mol Sci. 2022 May 16;23(10):5539." is an excellent review article that addresses precisely one of the themes that the authors consider to be promising – the development of immunotherapy for the treatment of endometriosis, through the blocking of NK cell inhibitors receptors. We made reference to it in our discussion.

The article "Freitag N, Baston-Buest DM, Kruessel JS, Markert UR, Fehm TN, Bielfeld AP. Eutopic endometrial immune profile of infertility-patients with and without endometriosis. J Immunol Reprod. 2022 Mar; 150:103489" evaluates eutopic endometrial expressions of several immune cells (uterine natural killer cells, plasma cells, macrophages and CXCL1) in infertility-patients with endometriosis. We consider that it is outside the scope of our review. We focused our work on NK cell receptors in the ectopic endometrium.

Regarding the article "Peng H, Weng L, Lei S, Hou S, Yang S, Li M, Zhao D. Hypoxia-hindered methylation of PTGIS in endometrial stromal cells accelerates endometriosis progression by inducing CD16- NK-cell differentiation.  Exp Mol Med. 2022 Jul;54(7):890-905", our review only includes human studies, and this study uses animal models. In addition, this study focuses on the evaluation of Prostacyclin (PGI2) as a promoter of endometriosis, which is also outside the scope of our review.

Reviewer 2 Report

It is a very interesting paper. 

NK have an important role in reproductive life and to try to understand e clarify any possibile role in endometriosis probably can represent a new strategy to control the disease, symptoms but also "side effects" such as infertility, repeated implantation failure, low ovarian reserve. Probably all this aspects have a link with the iummunological aspects of endometriosis. A new strategy to treat/control endometriosis in order to improve the quality of life of all the women affected. 

Some studies reports a medical treatment in patients enrolled (eg oral contraception pill). Probably it could be interesting to report if there are differences in the studies results in relation to this aspect. 

Author Response

Reply to Reviewer #2

The question of how hormonal therapy can influence the expression of NK cell receptors is an extremely relevant and interesting issue, so we thank the reviewer for having drawn attention to this topic. The authors also took this aspect into account and analyzed the included studies, however they found a great heterogeneity of criteria selection. Six studies made no reference to inclusions or exclusions of patients undergoing this type of therapy; two studies excluded women who had undergone GnRH analogues in the last 12 months; two studies excluded those who underwent GnRH analogues in the last three years (in both there is no reference to the contraceptive pill); six studies excluded patients who had undergone any hormonal therapy in the previous three months and in one study all those who had undergone any hormonal treatment in the last six months were excluded. With this criteria heterogeneity, it was not possible to determine or organize comparisons between the different studies. We believe, however, that it is an excellent topic for future research work in which this issue can be analyzed as a primary objective.

Round 2

Reviewer 1 Report

In my opinion, all necessary changes have been made for the article to appear in IJMS.